# Workplace Spirituality as an Alternative Model for Promoting Commitment to Change and Change-Oriented Organisational Citizenship Behaviour

**Sulastini [1,***], Tri Cicik Wijayanti [2] and Ismi Rajiani [3]**

[1] Magister Management Program, Universitas Islam Kalimantan Muhammad Arsyad Al Banjari, Banjarmasin 70123, Indonesia
[2] Magister Management Program, Universitas Gajayana Malang, Malang 65144, Indonesia
[3] Department of Social Studies, Lambung Mangkurat University, Banjarmasin 70123, Indonesia
[*] Correspondence: aisyahsulastini@gmail.com

**Abstract:** Considering that the relationship between workplace spirituality and organisational change has only recently sparked scholarly interest, the connection could be more evident, as research has only recently begun. Furthermore, there is a growing tendency to add spirituality as the fourth dimension to sustainability in addition to the environment, social responsibility, and the economy. This study contributes to the academic literature by examining the influence of workplace spirituality on lecturers' responses to change in their commitment to change and change-oriented organizational citizenship behaviour (C-OCB). Private universities in Banjarmasin, Indonesia, were selected to research these links. To collect data from 1050 instructors, self-administered questionnaires were used, and structural equation modelling (SEM) was performed. The findings demonstrated that the workplace spirituality level, exemplified by a solid connection to a higher power, humanity, and nature, affects the lecturers' commitment to change leading to change-oriented organizational citizenship behaviour (C-OCB), a dimension of OCB less researched. The findings offer a novel perspective on the relationship between workplace spirituality, lecturers' attitudes towards change, and change-oriented organizational citizenship behaviour (C-OCB) in higher education. Since this result has not been examined in the context of change in higher education, several consequences and suggestions for future research emerge. More studies could be used in future research, given that the literature in this area is still in its early stages. The current work is expected to spark future research on this intriguing and fertile issue.

**Keywords:** workplace spirituality; universities; lecturers; OCB; commitment to change; organisational change

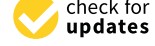



## 1. Introduction

Organizations currently face fierce competition both domestically and internationally from companies defined by tenacious effort, a dynamic environment, innovative strategy, and the present COVID-19 pandemic in a volatile, uncertain, complicated, and ambiguous world (Clauss et al. 2022). As a result, organisations are susceptible to continually developing new models to prepare for and manage erratic shifts. The education sector is also vulnerable due to ongoing pedagogical philosophy changes, technological advancements, and new methods of instruction delivery (Lyn Chan and Muthuveloo 2021). The glaring example is that during the COVID-19 pandemic, professors had to switch from traditional learning to an online platform (Lee and Jung 2021). These changes are attainable if the teaching staff devote themselves to change, as committed lecturers are capable of establishing a constructive attitude towards new difficulties in higher education (Moriña and Orozco 2022).

Indonesia's higher education system has experienced several revisions as a result of changes in the nation's economy, government, and society over the past 20 years. These improvements included university reform to boost leadership practises, learning outcomes, instructional techniques, and management innovation (Rajiani and Ismail 2019). Academics, however, continue to be essential to the success of university reform because of their vital role in turning vision into reality (Fatimah et al. 2022). The standard and reputation of universities are inextricably linked to the calibre of lecturers' instruction, which has a big impact on students' contentment and the long-term viability of the institution (Ali et al. 2021). The institution's administration is therefore required to retain the professors who demonstrate organisational citizenship behaviour (OCB), also known as pro-social behaviour, and those who are adaptable (Perryman and Calvert 2020).

Change-oriented organisational citizenship behaviour is a unique domain of OCB defined by Bettencourt (2004) as inventive and creative employee acts focused on bringing about positive organisational change. Later studies indicated that although OCB activities are essential, more is needed to ensure an organisation's sustainability (Jang 2021). Therefore, a business also requires people who are willing to challenge the current operational status to affect positive change (Chiaburu et al. 2022). Nowadays, this type of job performance is known as change-oriented OCB (C-OCB). In contrast to the plethora of research on assistance and compliance, this component of OCB has only received scant attention (Zampetakis 2022).

Employee commitment in the change management story denotes an attitude towards change (Al-Shamali et al. 2021). Although employers expect their staff to comply with change, the initiative's success depends on the staff members' willingness to go above and beyond the call of duty. Recognizing an employee's devotion is essential since it helps justify their desire to go above and beyond the requirements of their position (Ford et al. 2021). An employee's connection to new guidelines, procedures, plans, budgets, and technology is demonstrated by their commitment to change. For modern business organisations, affective commitment to change is preferred because this is the mindset associated with supporting change behaviours (Mihalache and Mihalache 2022). However, the influence of commitment orientations on individuals' change-readiness may vary depending on the country in which the study is conducted due to changes in national cultural characteristics between countries (Rajiani and Pypłacz 2018).

Although the Indonesian higher education system has publicised considerable growth and progress in refining quality education in the last two decades, the significant problems commonly disturbing Indonesian universities are unfulfilled missions and unsuccessful organisational objectives due to failures of governance at the institutional level (Rosser 2022). This is revealed by the fiasco of most Indonesian universities in competing with their counterparts across the southeast Asian region. In university rankings, most Indonesian universities are in an unsatisfactory position compared to the top universities from neighbouring Asian countries (Times Higher Education 2022). As a result of the severe competition among academics to increase the number of articles in high-indexed journals, organisations are searching for other techniques to solve concerns such as plagiarism, ghost authors, falsification, and citation farming. Such a need for alternative administration methods is also fuelled by the realisation that traditional initiative based on bureaucratic and mechanistic practises is no longer successful. Furthermore, the remarkable growth of some Asian countries in introducing different approaches to dealing with organisational problems has provided an opportunity to ascend to new patterns in the organisational field—even though they still practise nepotism as preferential treatment—with greater receptivity to spirituality (Iqbal and Ahmad 2020). Therefore, it is evident that Indonesian universities urgently need to reform their human resource management.

Since interest in examining the impact of workplace spirituality (WS) in the workplace has grown recently (Widyanti and Basuki 2021), scholars have begun to examine it in university environments to deal with dynamic changes and developments (Hadi et al. 2022). In addition, there is a growing understanding that spirituality, in addition to the

environment, social responsibility, and economic issues, should be seen as a fourth element of sustainability (Mohd Zawawi and Abd Wahab 2019; Luetz et al. 2023; De Guimarães et al. 2020). Hence, analysing WS can help Indonesian higher education systems, which are dealing with a number of issues influencing universities' competitiveness and academic results. Since there have been calls for research to examine the relationship between commitment and resistance to change (Kayani et al. 2022), the current study aims to examine how WS in Indonesian universities may affect C-OCB and commitment to change so that it can potentially be applied as an alternative model to improve lecturers' performance.

In Indonesia, where mostly Muslims reside, spirituality is frequently associated with religiosity. Specifically, this research was conducted in Banjarmasin, where Islam is a crucial component of ethnic recognition. As a way of life, all Banjarese Indonesians subscribe to Islam. Thus, Islamic ideals and behaviour pervade all spheres of experience (Basuki et al. 2021). In light of the fact that Indonesia is a Muslim nation with a secular constitution and institutional structure, it is essential to collect Indonesian samples. The dynamics of civil society have been a key factor in the country's spiritual growth. Thus, workplace spirituality derived from Islamic ethos is anticipated to be greater among Indonesian employees than among a sample of employees recruited from countries where Islam is limited to symbolic values or politicised. In spite of the recent resurgence of conservative Islam (Schäfer 2019), Indonesian Islam has historically been viewed as tolerant or moderate (Pektas 2021). There is no single religious practise that exemplifies Islam in Indonesia, resulting in a multiplicity of Islam's forms. In terms of reading and understanding the Qur'an, Islam's theology has diverged since its inception. As a result of this diversity, numerous schools of thought (*madhab*) have developed, ranging from the Sunni sects of Hanafi, Shafi'i, Maliki, and Hanbali to the Shi'a and Ahmadiyah. In addition, when the Islamic religion was declared and expanded, it had to battle with indigenous traditions. Hence, Islam in Indonesia is becoming increasingly syncretistic (Hefner 2020).

*Theoretical Development and Hypotheses*

Over the past twenty years, the study of WS has emerged as a new subfield within the disciplines of management and economics (Yin and Mahrous 2022). The World Bank, AT&T, Hewlett-Packard, DuPont, Ford Motor Company, Microsoft, Google, and Apple are just a few well-known companies that have built workplace spirituality to achieve superior performance (Garg 2020). After Toffler (1980)—the third technological wave—this points to the emergence of the fourth, spiritually based firm (Fraya and James 1999). The spiritual paradigm acknowledges that people work not just with their hands but also with their hearts or spirits (Paul et al. 2020).

Spirituality in the workplace is not necessarily about religion or converting others to a particular belief system. The fundamental distinction between the two is that religion has laws and particular beliefs to follow and is concerned with the sacred. In contrast, spirituality concerns the philosophy of life, attitudes, and values (Mohd Zawawi and Abd Wahab 2019). The concept of spirituality focuses on employees who perceive themselves as spiritual beings whose souls require nourishment at work; who experience a sense of purpose and meaning in their work; and who feel connected to their workplace community (Palmer Kelly et al. 2020). Spirituality at work recognises the inner life nurtured and nourished by meaningful labour in the community context. They also indicate that a genuinely learning firm with a comprehensive awareness of the employee will foster the development of an individual's IQ and emotional intelligence, as well as their soul (Zappalà 2021). Although there is no widely accepted definition of workplace spirituality (Saxena and Prasad 2022), there is an emerging consensus that spirituality is a multifaceted concept concerned with finding a connection to something meaningful that transcends our everyday lives (Jastrzębski 2022). Organisations with a more incredible feeling of spirituality in the workplace do better than those with little or no spirituality. In addition, organisations with a high level of spirituality expand faster, become more efficient, and have more excellent rates of return (Garg 2020; Garg et al. 2022; Al-Mahdy et al. 2022).

OCB does not result in formal rewards or prizes (Oplatka and Hassan 2021; Lin and Chi 2022) because the environment facilitates this behaviour by affecting employee emotions (Liu and Keller 2021). When universities embrace this perspective, faculty members' extracurricular and pro-social commitments rise, indicating a favourable and conducive environment for lecturers. OCB includes aiding colleagues, university management, and students in higher education. These include helping other lecturers prepare student activities, constructively discussing universities with the public, and communicating the university's vision, mission, and goals (Ghasemy and Elwood 2022). OCB improves lecturers' self-fulfilment and job happiness, students' academic performance, and the university's reputation, according to several experts (Garg et al. 2022; Manzoor et al. 2021). In collectivist Indonesia, OCB is important (Rajiani and Kot 2020), as collectivists advocate group welfare and help group members and organisations altruistically (Cohen and Abedallah 2021).

The dimensions of workplace spirituality: interconnection with a higher power, interconnection with human beings, and nature and all living things (Liu and Robertson 2011), are crucial aspects that motivate employees to perform OCB. People are connected not only when they connect with themselves through introspection and a deep awareness of their inner selves and when they combine different parts of themselves into a coherent whole, but also when they expand their boundary, the individual self, to include other people into the self and go beyond the categories of "us" and "them" to achieve harmony (Bayani and Serajzadeh 2022). Following this definition, employees with a strong transcendental self-identity are more likely to exhibit pro-social behaviour, which is the willingness to assist, protect, or promote the welfare of others (Garg et al. 2022). Therefore, this pro-social conduct that individuals exhibit at work can drive them to contribute beyond their regular obligations to aid their co-workers and the organisation.

Employees' spiritual experiences in the workplace were referred to as workplace spirituality (Hassan et al. 2022). When lecturers can express their desire for care and compassion for others, experience inner consciousness in the search for meaningful work, and achieve transcendence, it can be claimed that employees have a positive working experience. This will encourage individuals to enjoy their work and go above and beyond their commitments (job description) in the workplace (Margaretha et al. 2021). Thus, research indicates that workplace spirituality contributes to increased organisational citizenship behaviour. Affective event theory (AET) explains workplace spirituality's effect on organisational citizenship behaviour. According to AET, an event is a proxy that induces emotive emotions (Yang et al. 2022). This pleasant experience inspires individuals to enjoy their work and go beyond their commitments (job description) in the workplace. Thus, we hypothesise:

**H1.** *In Indonesian higher education, workplace spirituality positively relates to change-oriented OCB.*

Lecturer commitment is a lecturer's dedication to university leadership, student learning, professional growth, and university effectiveness (Abboh et al. 2022). Herscovitch and Meyer (2002) defined a commitment to change as an attitude that leads to the actions needed to undertake a change programme. This mentality shows (a) an aspiration to support the change because of a conviction in its advantages (affective commitment), (b) a recognition that failing to support the change has costs (continuance commitment), and (c) a sense of duty to support the change (normative commitment to the change). Thus, people who commit to a change because they value it (affective) or feel obligated (normative) should not only comply but go above and above by taking on additional tasks (Nixon and Scullion 2022). However, individuals who understand the consequences of not supporting a change (continuance) will only do what is required (Min and Ye 2021).

Affective commitment among academic employees is enhanced when the organisation respects and supports them. Affective commitment is reinforced and demonstrated through social contacts, social recognition, and the day-to-day operation of social capital in the workplace. When personnel internalise the university's norms and ideals through day-to-day socialisation and involvement, normative commitment develops and is instilled. Staff

gain specific benefits from workplace socialisation, some concrete and some intangible that lead them to feel the need to reciprocate and absorb the organisation's values and traditions. The emphasis of continuance commitment is on workers understanding the expenses of staying or leaving the university and basing their commitment on this calculation (Margaretha et al. 2021).

Organisational change management requires employee commitment to change, and innovation only occurs when people create and implement ideas. Due to organisations' seemingly unending efforts to re-engineer their business processes to compete, employee readiness for change has become a key concern. This is especially relevant given the many strategic change deployments that failed due to employee disengagement and involvement (Alqudah et al. 2022). Employee commitment is critical to successful organisational transformation (Tipu 2022). To respond to issues faster and capitalise on emerging market opportunities, firms should match their organisational structure, policies, and strategies by integrating and nurturing change and innovation. Organisational culture, social ties, management/leader relationships, job knowledge and abilities, job expectations, ability to cope with change, colleague support, moral beliefs, and attachment to the organisation might affect employees' willingness to change (Potosky and Azan 2023). Spirituality is seen as having the capacity to contribute to organisational stability and sustainability by affecting affective and normative organisational commitments (Aboobaker 2022). However, affective commitment to change (CC) was chosen as the sole commitment scale for use in this study since it is the best criterion for organisational commitment in an Asian environment (Lau et al. 2016).

Thus, we hypothesise:

**H2.** *Within the Indonesian higher education context, workplace spirituality is positively related to lecturers' affective commitment to change.*

Emirie and Gebremeskel (2022) emphasise that organisational commitment has a significant effect on organisational citizenship conduct in an educational setting. Because the affective, continuance, and normative types of commitment differ in character, their impact on readiness for change may differ. According to a prior study, employees with a strong affective commitment believe in the change and want to help in its achievement (Alqudah et al. 2022). In our theoretical model, we chose to focus on affective organisational commitment because it has been shown to predict OCB more strongly than other dimensions of commitment and captures the social exchange processes that occur when employees decide to reciprocate their organisation's favourable treatment (Schwarz et al. 2023). However, in order to boost performance, there is more needed than merely building a happy environment and employing hard-working workers. Having cooperative and obedient personnel can be detrimental to a business: "a worker who goes above and beyond the call of duty to complete a misconceived task may be more damaging to an organisation than a more humdrum performer" (Ren et al. 2022). With rising competitiveness and unpredictability in the business environment, employees are expected to be more proactive, adaptable, and creative in task-related interactions. Change-oriented organisational citizenship behaviours are proactive employee actions intended to identify and execute improvements in products, services, or work procedures. Change-oriented OCB is comparable to classic citizenship OCB in that all of these activities are viewed as going above and beyond an employee's statutory job responsibilities. They differ in that affiliated types of classic OCB emphasise assisting others and the business with the maintenance of existing processes and procedures, whereas change-oriented OCB focuses on enhancing them. Positive, proactive behaviour, creative performance, voice (speaking out with change proposals), a personal initiative to tackle work difficulties, adaptable performance, and taking charge is typical change-oriented OCB (Chiaburu et al. 2022; Bettencourt 2004). As a result of the initial investigation, the following hypotheses are offered in this study:

**H3.** *In Indonesian higher education, affective commitment to change positively relates to lecturers' change-oriented OCB.*

**H4.** *Within the Indonesian higher education context, affective commitment to change mediates the relationship between workplace spirituality and change-oriented OCB among lecturers.*

The independent, dependent, and mediating variables of this study are shown in Figure 1.

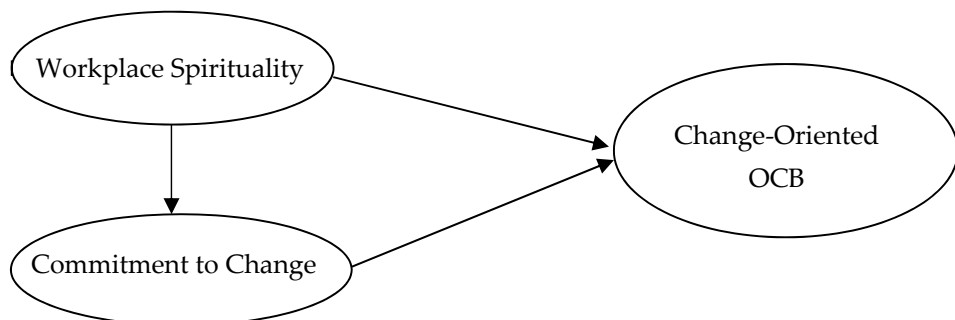

**Figure 1.** The relationship between concepts.

## 2. Results

Five hundred and twenty-eight (50.3%) respondents were male, and 522 (49.7%) were female. With regard to academic position, the most prevalent was lecturers, 735 (70%), followed by assistant professors at around 158 (15%), associate professors at approximately 147 (14%), and finally, professors at approximately 10 (1%). Regarding age group, 231 (22.2%) respondents were between 20 and 30 years old. Approximately 476 (45.3%) respondents were between the ages of 31 and 40. Two hundred and fifty-six (24.4%) responders were between 41 and 50. The remaining 75 (7.1%) responders were aged 51 and older. In terms of working experience, 258 (24.6%) respondents had served at the universities for one to three years, 304 (29.0%) respondents had served between four and six years, 202 (19.2%) respondents had served between seven and ten years, and the remaining 285 (27.2%) respondents had served at the universities for more than eleven years. In terms of academic credentials, 755 (71.9%) respondents held a master's degree and 295 (28.1%) had a doctorate. The mean of each variable is displayed in Table 1. Discerning the mean score of workplace spirituality, 4.5, the typical respondent for this research is considered to have high interconnection with a higher power, interconnection with human beings, nature and all living things. The mean score of commitment to change, 3, denotes that the people in the organisation are a combination of those who are ready and committed to change and those who are still in doubt about embracing the change. The mean score of change-oriented OCB, 2, signifies weak intention to rectify a flawed procedure, an erroneous job description, or an unrealistic role expectation.

**Table 1.** Variable means.

| Variables | N | Mean | Std. Error |
|---|---|---|---|
| Workplace spirituality | 1050 | 4.5 | 0.402 |
| Commitment to change | 1050 | 3.2 | 0.302 |
| Change-oriented OCB | 1050 | 2.1 | 0.231 |

In describing the outcomes, we discuss the model's validity and the hypotheses. The validity of the estimation model is determined by ensuring that all constructs have factor loading values greater than or equal to 0.5 (50%) (Hair et al. 2020). As shown in Table 2, all values for the constructs whose values surpassed 65% are acceptable.

**Table 2.** Measurement model.

| Construct | Items | Loading Factors |
|---|---|---|
| Workplace spirituality | Interconnection with a higher power:<br>1.    I believe that death is a doorway to another plane of existence.<br>Interconnection with human beings:<br>2.    I am quickly and deeply touched when I see human misery and suffering.<br>Interconnection with nature and all living things:<br>3.    My life is intimately tied to all of humankind. | 0.824<br><br>0.810<br><br>0.734 |
| Affective commitment to change | 1.    I believe in the value of this change.<br>2.    This change is a good strategy for my university.<br>3.    Management is making the right decision by introducing this change.<br>4.    This change serves an essential purpose.<br>5.    My university would be better with this change.<br>6.    This change is necessary. | 0.752<br>0.805<br>0.643<br>0.701<br>0.755<br>0.712 |
| Change-oriented OCB | 1.    I adopt more efficient methods for performing my duties.<br>2.    I alter the manner in which the job is performed to make it more efficient.<br>3.    I am attempting to enhance the organisation's practises.<br>4.    I implement new, more efficient work practices for the organisation.<br>5.    I provide positive suggestions for enhancing the organisation's operations.<br>6.    I attempt to rectify flawed methods or practices.<br>7.    I attempt to reduce redundant or superfluous procedures.<br>8.    I attempt to execute solutions for urgent organisational issues.<br>9.    I make an effort to implement innovative methods of work to boost productivity. | 0.928<br>0.952<br>0.903<br>0.879<br>0.802<br>0.917<br>0.902<br>0.762<br>0.632 |

The model provided in Figure 2 also suited the data well. According to Shipley and Douma (2020), this model meets the model's goodness-of-fit by referring to Chi-square ($\chi^2$) = 132.212 (less than 639,232); and probability ($p$ = 0.05). Moreover, by referring to Hair et al. (2020), the model displayed good fitness: CMIN/DF = 1. 29 (expected smaller than 2), GFI = 0.991 (exceeding 0.90), CFI = 0.951 (exceeding 0.95).

After fitting the data, the structural model was examined. The goal of the structural model was to accept or reject the hypotheses and highlight the nature of the shared relationships among the investigated variables. As shown in Figure 2, workplace spirituality has a strong, significant, and direct effect on affective commitment to change (=0.70, $p$-value = 0.01) and a moderate, significant, direct effect on organisational citizenship behaviour (=0.51, $p$-value = 0.02). Moreover, affective commitment to change had a moderately significant direct effect on C-OCB (=0.33, $p$ = 0.01). Thus, the SEM results confirm the first three hypotheses of the investigation and provide preliminary support for the mediating role.

The summary result of structural equation modelling to test the hypotheses is presented in Table 3.

**Table 3.** Path relationship among constructs.

| Hypothesis | Influence | Estimate | CR | *p* | Conclusion |
|---|---|---|---|---|---|
| | | Direct Effect | | | |
| H1 | WS → C-OCB | 0.70 | 12.324 | 0.01 | Supported |
| H2 | WS → CC | 0.51 | 8.765 | 0.02 | Supported |
| H3 | CC → C-OCB | 0.33 | 6.421 | 0.01 | Supported |
| | | Indirect Effect | | | |
| H4 | WFE → JS → TI | 0.70 × 0.51 × 0.33 = | | 0.118 | Supported |

Since this study employed an implicit method to test mediation (Preacher and Hayes 2008), the positive path of WFE → JS → TI = 0.70 × 0.51 × 0.33 = 0.118 implicitly ac-

cepts the fourth hypothesis that within the Indonesian higher education context, affective commitment to change mediates the relationship between workplace spirituality and change-oriented OCB among lecturers.

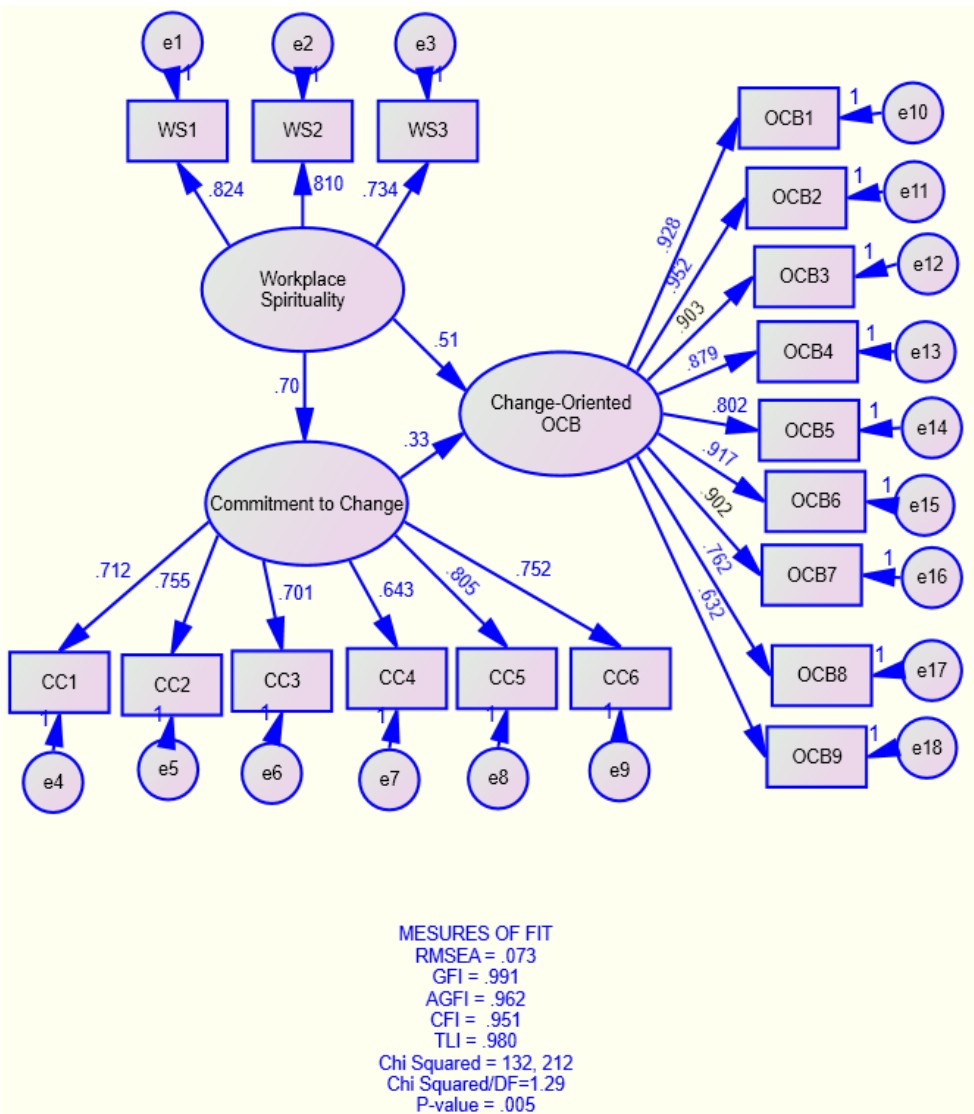

**Figure 2.** Results of the SEM analysis of the conceptual model.

The study results indicate that WS and affective commitment to change are crucial characteristics that contribute to OCB. The sequence of studies provides additional support for the efficacy of the mediation hypothesis, suggesting that WS contributes directly and indirectly to change-oriented OCB by strengthening lecturers' affective commitment to change. This study's fourth hypothesis demonstrated how the intervening variable (commitment to affective change) influences the link between WS and OCB.

## 3. Discussion

In previous studies from higher education institutions (Margaretha et al. 2021; Cohen and Abedallah 2021; Manzoor et al. 2021; Oplatka and Hassan 2021; Al-Mahdy et al. 2022; Garg et al. 2022) and business in general (Ford et al. 2021; Ghasemy and Elwood 2022; Kayani et al. 2022), the central study question was the impact of spirituality on various organisational outcomes, and as previously said, few studies have looked into the relationship between workplace spirituality, commitment to change, and change-oriented OCB. The primary purpose of this study was to determine if workplace spirituality stimulates

change-oriented organisational citizenship behaviour (OCB) among employees, especially given the mediating function of their commitment to change. The hypothesised model was consistent with the data, with various elements of workplace spirituality influencing change-oriented organisational citizenship behaviour. According to this study's hypothesis, employees with greater interconnection with a higher power, human beings, and nature demonstrated more outstanding commitment to change and change-oriented OCB. When individuals perceive their work as essential and purposeful, they will want to add value to it by engaging in extra-role behaviours, as was the case with change-oriented OCB in this study (Chiaburu et al. 2022; Bettencourt 2004). Following previous research demonstrating that workplace spirituality has a significant impact on encouraging employees' flexibility and commitment to change, our study demonstrated that employees' exposure to workplace spirituality had a considerably favourable impact on their change-oriented OCB (Hassan et al. 2022; Margaretha et al. 2021). Open, informal, and personal ties with co-workers create conditions conducive to igniting and encouraging employee change behaviours. Employees who think their immediate work community is receptive to fresh ideas and proposals enjoy greater job satisfaction. They are more likely to demonstrate greater openness and adaptability to change (Alqudah et al. 2022). Sense of closeness and togetherness in the workplace reduces employees' apprehensions regarding re-engineering and change processes, impacting their readiness and adaptability to changes, and hence change-oriented organisational citizenship behaviour (Schwarz et al. 2023). Although this study's findings demonstrated a significant direct relationship between spirituality and lecturers' commitment to change and change-oriented OCB, the mean of the latter variable was rated the lowest (2.1 out of 5). This conclusion may reflect the unique work environment in current educational sectors during the COVID-19 pandemic, when the rate of volatility, uncertainty, complexity, and ambiguity (VUCA) is relatively high. This may be different in other industries, which requires additional research.

An organisation's vision, mission, and objectives provide a long-term perspective on the extent to which management pushes for innovation and preferable employee behaviour. To align individual and employee aims and values in this university sample, managers must concentrate on individual behaviours (spirituality, commitment to change, change-oriented OCB), team functioning, and organisational processes. For transformation to occur, corporate processes, systems, structures, and culture must all be aligned with employee and organisational viewpoints (Potosky and Azan 2023). When implementing organisational transformation, leaders must establish and align activities that distinguish the organisation from its competitors and foster its competitive advantage at the individual, team, and organisational levels. As the level of spirituality is high, universities can distinguish themselves as spiritually-based institutions (Fraya and James 1999). This has implications for the management paradigm, which recognises that people work not only with their hands but also with their hearts or spirits (Paul et al. 2020). Lecturers bring their bodies, minds, and souls to work. In this approach, the spiritual component of people in the workplace unlocks lecturers' full creativity and potential, while employees develop as whole individuals.

Change management programmes are only effective if managers link employees' passions and energy with the initiative's objectives. In the context of a university's commitment to change, this is one of the first studies to evaluate the influence of workplace spirituality on change-oriented organisational citizenship behaviour. Theoretically, studying the antecedents of change commitment helps researchers comprehend and prepare for the organisational change procedure. Universities should instead focus on building a workplace culture and atmosphere that encourages employees to produce creative ideas (Rajiani and Ismail 2019) in light of the low productivity of lecturers, concerns about the quality of outputs, and dwindling enrolment. This study provides empirical evidence that focusing on employees' personal beliefs, growth, and development can positively impact employee attitudes and, eventually, the entire organisation. To encourage employee participation

in extra-role activities, management should develop platforms for open dialogue about employees' opinions, values, and rights and combine organisational and personal goals.

This study's findings also imply that managers should seek to assist employees in recognising the intrinsic value of their work and, as a result, generate a better sense of purpose in their employment.

Universities are faced with the challenge of managing employees whose values are compatible with the organisation's ideals as organisations become increasingly diverse due to differing backgrounds, cultures, nations, and values. Influential university leaders who create and sustain working environments characterised by spiritual values, such as openness, ethics, and acceptance of varied opinions, are more likely to motivate their staff to engage in extra-role behaviours. These activities include adopting more effective techniques for completing responsibilities and implementing creative work methods to increase productivity. Managers must be aware of the factors that generate change-oriented organizational citizenship behaviour (OCB) among their employees. Organisations must cultivate spiritual traits such as trust, acceptance, and justice within their organisational cultures. A more profound sense of community in the workplace and strong ties with each other, customers, and the community may assist employees in strengthening interpersonal interactions and gaining support for their ideas.

In general, all Banjarese Muslims in the samples are aware that Islam is a way of life, which must be reflected in all facets of their lives, including the job. Specifically, Islam teaches its followers that their responsibilities and work are elements of worship. Regrettably, the work environment is not always as expected. There are times when believers may face challenges at work that might lead to wrath, discontentment, anxiety, jealousy, stress, and even burnout. If these factors are not taken into account and rectified, the commitment to change and OCB are severely impacted. Previous research has indicated that workplace spirituality has a high propensity and capacity for motivating employees to efficiently carry out their obligations (Ford et al. 2021; Ghasemy and Elwood 2022; Kayani et al. 2022). OCB is the consequence of a commitment to change. However, the majority of prior research on the association between workplace spirituality and OCB aimed to expand the current body of information. By analysing the link between WPS, CC, and OCB within the context of private Indonesian higher education institutions, this study expands the current corpus of related knowledge.

Change-oriented organizational citizenship behaviour provides empirical support for various theoretical ideas regarding the relative contributions of these categories to the understanding of organisational outcomes supporting previous studies (Margaretha et al. 2021; Cohen and Abedallah 2021; Manzoor et al. 2021; Oplatka and Hassan 2021; Al-Mahdy et al. 2022; Garg et al. 2022; Ford et al. 2021; Ghasemy and Elwood 2022; Kayani et al. 2022). Among all models, workplace spirituality showed the greatest explanatory value for predicting change-oriented organizational citizenship behaviour. This effect was anticipated based on the notion that organisational behaviour and attitudes are best comprehended as being motivated by the interaction between the individual and the commitment. Consequently, this study contributes to the expanding discourse on workplace spirituality as an indirect vs direct predictor of change-oriented organisational citizenship behaviour. Lecturers frequently cite intrinsic motivations, such as assisting and serving others, as their driving force. It is crucial to understand the relationship between workplace spirituality, commitment to change, and change-oriented organisational citizenship behaviour, given that the current societal climate may impair these characteristics.

To sum up, universities that face unpredictable business contexts marked by change and disruption must embrace innovative HR policies, initiatives, and strategies. Since workplace spirituality has emerged as an innovative organisational practice, especially in the context of a paradigm shift among lecturers seeking more meaning and purpose in their work than just materialistic outcomes, this model can be used as an alternative to foster employee attitudes and behaviours in terms of change readiness, resulting in change-oriented OCB. The desire of lecturers to change drives a university's ability to adapt

to emerging advancements and innovative ideas. As a result, university executives face the difficulty of creating an atmosphere where employees are encouraged to engage in extra-role behaviours that contribute to the organisation's performance.

## 4. Materials and Method

This study utilised quantitative methodology to investigate the associations between workplace spirituality, OCB, and affective organisational commitment to change. The research was conducted from June 2022 until January 2023. The targeted population was all lecturers who worked at private universities in Banjarmasin, Indonesia numbering around 1500 lecturers. From this total population, random sampling was employed. Only 70% (n = 1050) of the total lecturers agreed to participate in the study. The number of respondents was sufficient to meet the minimum recommended number of cases (n = 200) for structural equation modelling. All participants had a minimum of three years of teaching experience and a maximum of 25 years. WS was taken from the 3-item scale developed by Liu and Robertson (2011) and labelled as interconnection with a higher power, and interconnection with human beings, nature, and all living things. Affective commitment to change was measured using adapted 6-item scales that Herscovitch and Meyer (2002) developed. Change-oriented OCB was measured using adapted 9-item scales by Chiaburu et al. (2022). All the items were assessed on a five-point Likert scale ranging from 1 (strongly disagree) to 5 (strongly agree), where a higher rating indicated greater strength or presence of the construct. Structural equation modelling was utilised since this methodology was developed to validate substantive theory using empirical data. In this study, the theory predicts that certain aspects of workplace spirituality do not influence other dimensions and that certain dimensions of organisational citizenship behaviour do not load on specific elements; hence, SEM was best suited to test the theory. SEM consists of a series of statistical processes that permit the evaluation of causal relationships between latent variables and a set of observable variables. The correlations or effects presented by the model are supported by a suitable, comprehensive measurement. The estimation model's validity is performed by observing confirmatory factor analysis (CFA). We removed items that failed to achieve the desired cutoff criterion with a factor loading of 0.60 (Hair et al. 2020). We used the bootstrapping analysis method to examine indirect effects in the structural model (Preacher and Hayes 2008). The parameter estimates applied several fit indices: chi-square/degrees of freedom ratio ($\chi^2$/df), comparative fit index (CFI), normative fit index (NFI), Tucker–Lewis index (TLI), and root mean square error of approximation (RMSEA). CFI, NFI, and TLI should be $\geq$0.90; RMSEA should be <0.08, and the $\chi^2$/df value must be < 0.3 for the criteria of a good model fit (Hair et al. 2020).

## 5. Conclusions

This study aimed to empirically investigate the differential predictive capacities of connectivity with a higher power, humanity, and nature with organisational principles to understand better employees' adaptation to change and change-oriented organisational citizenship behaviour. The validation of the theoretical framework proposed in this study contributes to the organisational literature's efforts to understand the role of workplace spirituality in influencing lecturers' commitment to change and change-oriented organisational citizenship behaviour. Finally, the current study suggests that organisational efforts should create a work environment that promotes employees' social and psychological requirements for connectedness, inclusion, influence, and attachment. Such positive experiences of connectedness have a constructive effect on their job attitudes and outcomes, which benefits both the individual and the business and supports them in achieving better effectiveness and sustainability in today's uncertain economic situation. However, given that the study was conducted among education sector professionals, the conclusions may need to be revised in their relevance to other industries. Despite these limitations, this study makes an essential contribution to the literature on organisational behaviour as the first to investigate the relationships between workplace spirituality, employee commitment

to change, and change-oriented organisational citizenship behaviour (OCB). It substantiates the importance of fostering spirituality in the workplace and shows how it might promote excellent job outcomes and extracurricular behaviour among academics.

**Author Contributions:** Conceptualization, S. and I.R.; methodology, I.R.; software, I.R.; validation, T.C.W., S. and I.R.; formal analysis, S.; investigation, T.C.W.; resources, S.; data curation, T.C.W.; writing—original draft preparation, I.R.; writing—review and editing, S.; visualization, T.C.W.; supervision, S,; project administration, I.R.; funding acquisition, T.C.W. All authors have read and agreed to the published version of the manuscript.

**Funding:** This research received no external funding.

**Institutional Review Board Statement:** Not applicable.

**Informed Consent Statement:** Not applicable.

**Data Availability Statement:** Available upon request.

**Conflicts of Interest:** The authors declare no conflict of interest.

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
