# Peer review of "Workplace Spirituality as an Alternative Model for Promoting Commitment to Change and Change-Oriented Organisational Citizenship Behaviour"

_admsci, doi:10.3390/admsci13030086_

Round 1

Reviewer 1 Report

(1). Abstract: Reads well

(2). Keywords: Add one more appropriate keyword

(3). Introduction: Reads well

(4). Literature review: Incorporate and apply systematic literature review approach

(5). Materials and Method: Elaborate the details of the research methodology. Justify, why adopted the SEM approach. Discus the suitability of this methodology for this problem. Address the advantages of this methodology over other methodologies.

(6). Results: Add more interpretations. Highlights the novel aspects

(7). Discussion: Explained well

(8). Theoretical and practical implications of the findings: Incorporate theoretical and practical implications

(9). Conclusion: Add limitations of this study

(10). Citations/references are not proper format in the entire paper. It should be corrected. Attention should be paid to clarity of expression and readability

Reviewer 2 Report

The paper reads very well and discusses an interesting topic. I think it is worthy of publication. English language is clear and academic too. The paper has been well cited as well. I have only two main comments of concern however. First, you discussed the issue of spirituality very well but did not discuss equally how this differs from the characteristics of a secular workplace. You did not discuss in an adequate manner the advantages of secular workplace settings and how these settings also provide or perhaps better provide the conditions supporting change. This is expected to underpin the argument and the proposition of the paper. Second, although the paper is discussing the Indonesian experience, it should also draw some comparisons with other countries which could have similar, better, or perhaps poorer experiences when it comes to applying this concept of spirituality to workplace. I am asking here just to draw some comparisons with other countries, not to fully analyse their data. Perhaps comparing them to other wide-spread religions, such as Christianity, Hindu, or Buddhism which have huge impacts on their believers, who can also apply the concept of spirituality to their workplace settings. 

Reviewer 3 Report

The paper's idea is well in time, and it has the potential to contribute. This interesting study investigates an exciting and important issue “Workplace Spirituality as an Alternative Model for Promoting Commitment to Change and Change-Oriented Organisational Citizenship Behavior”. However, some areas need further improvement.

1.     The title reflects the intention and the article's topic quite well. The abstract is written very informatively, but it does not get the reader's attention. The author/s could do more to explain what methodology is (data collection, sample, sampling technique, research tools) and highlights the study's contribution.

2.     Introduction part could be improved. To make this article more readable, the author/s can divide the current introduction into two parts. The first heading introduction and the second heading could be “Theoretical Background and Research Hypothesis”.

3.     The introduction is still a description and does not contain critical and in-depth discussion. The linkage between the manuscript's objective and why the research was necessary was not well established. The style of introduction must be coherent, and it should explain what the problem is, what has been researched in previous academic literature in this area, and what actual gap exists. Moreover, how this study fulfills this research gap. I  suggest revising the introduction part and providing what has been researched on this topic and the research gap.

4.     The article does not contain a critical literature review to support and justify the research gap. I suggest the author/s add a critical literature review and cite some latest literature sources. On lines 118-124, the author/s talked about the concept of spirituality. I think this conception of spirituality is closer to the concept of Alister Hardy.

·       Hardy, A. (1981). The spiritual nature of man. A study of contemporary religious experience. Tijdschrift Voor Filosofie, 43(3). https://philpapers.org/rec/HARTSN

5.     Please state the dimensions of spirituality that you have considered for this study and explain why you have chosen those dimensions.

6.     Consider reading the following research papers to improve your literature review section.

·       Workplace spirituality and organizational citizenship behavior: Evidence from banking industry. Management Science Letters, 4(8), 1685-1692.

·       The contribution of workplace spirituality on organizational citizenship behavior. Advances in business research, 6(1), 32-45.

 https://journals.sfu.ca/abr/index.php/abr/article/view/67

7.     Author/s can move the “Material and Methods” before analysis of results and discussion.

8.     The methodology used for the study is acceptable but suffers some minor limitations. More importantly, the choice of variables should be explained in light of the theory and the prior literature on the topic. The authors need to provide some more references to prior studies measuring similar theoretical constructs. Results are discussed appropriately but they could be improved. Most importantly, it is not clear which results contribute to which theoretical stream and how you position your study's results, in general, the existing studies.

9.      Author/s described the summary of the results in the conclusion part. I suggest author/s may rewrite the conclusion part. It must start by explaining the purpose and what has been done previously in the domain of studies. Also, provide future research directions precisely in the respective domain and suggest practical recommendations for society and academics.

10.   Practical implication needs to be strengthened.

Round 2

Reviewer 1 Report

Theoretical and practical implications of the findings: Incorporate theoretical and practical implications

Reviewer 3 Report

The paper's idea is well in time, and it has the potential to contribute. This interesting study investigates an exciting and important issue “Workplace Spirituality as an Alternative Model for Promoting Commitment to Change and Change-Oriented Organisational Citizenship Behavior”.

The author has incorporated the required changes. I suggest that the Author/s can move the “Material and Methods” before the analysis of results and discussion.
